# Machine Learning Methods for Gene Selection in Uveal Melanoma

**DOI:** 10.3390/ijms25031796

**Published:** 2024-02-01

**Authors:** Francesco Reggiani, Zeinab El Rashed, Mariangela Petito, Max Pfeffer, Anna Morabito, Enrica Teresa Tanda, Francesco Spagnolo, Michela Croce, Ulrich Pfeffer, Adriana Amaro

**Affiliations:** 1Laboratory of Gene Expression Regulation, IRCCS Ospedale Policlinico San Martino, 16132 Genova, Italy; francesco.reggiani@hsanmartino.it (F.R.); mariangela.petito@hsanmartino.it (M.P.); anna.morabito@hsanmartino.it (A.M.);; 2Department of Experimental Medicine (DIMES), University of Genova, Via Leon Battista Alberti, 16132 Genova, Italy; 3Institute of Numerical and Applied Mathematics, University of Göttingen, 37083 Göttingen, Germany; m.pfeffer@math.uni-goettingen.de; 4Skin Cancer Unit, IRCCS Ospedale Policlinico San Martino, 16132 Genova, Italy; enricateresa.tanda@hsanmartino.it (E.T.T.); francesco.spagnolo@jhsanmartino.it (F.S.); 5Department of Internal Medicine and Medical Specialties, University of Genova, Viale Benedetto XV, 16132 Genova, Italy; 6Department of Surgical Sciences and Integrated Diagnostics (DISC), University of Genova, 16132 Genova, Italy; 7Biotherapies, IRCCS Ospedale Policlinico San Martino, 16132 Genova, Italy; michela.croce@hsanmartino.it

**Keywords:** uveal melanoma, multi-domain data, data fusion

## Abstract

Uveal melanoma (UM) is the most common primary intraocular malignancy with a limited five-year survival for metastatic patients. Limited therapeutic treatments are currently available for metastatic disease, even if the genomics of this tumor has been deeply studied using next-generation sequencing (NGS) and functional experiments. The profound knowledge of the molecular features that characterize this tumor has not led to the development of efficacious therapies, and the survival of metastatic patients has not changed for decades. Several bioinformatics methods have been applied to mine NGS tumor data in order to unveil tumor biology and detect possible molecular targets for new therapies. Each application can be single domain based while others are more focused on data integration from multiple genomics domains (as gene expression and methylation data). Examples of single domain approaches include differentially expressed gene (DEG) analysis on gene expression data with statistical methods such as SAM (significance analysis of microarray) or gene prioritization with complex algorithms such as deep learning. Data fusion or integration methods merge multiple domains of information to define new clusters of patients or to detect relevant genes, according to multiple NGS data. In this work, we compare different strategies to detect relevant genes for metastatic disease prediction in the TCGA uveal melanoma (UVM) dataset. Detected targets are validated with multi-gene score analysis on a larger UM microarray dataset.

## 1. Introduction

In the last two decades, next-generation sequencing data have unveiled the genomics behind cancer and genetic diseases at a high level of detail. A consistent amount of this data has been made available in public databases related to projects such as The Cancer Genome Atlas (TCGA), the Personal Genome Project or repositories such as the European Genome-phenome Archive [1,2,3].

Public repositories of NGS data store processed or raw datasets, while in the first case data are directly available for bioinformatic analysis, as an expression matrix with samples in columns and genes in rows. In the other case, data should be preprocessed and prepared for further analysis. Several ad hoc public pipelines to obtain human readable files (as expression matrices or variant call format) from raw data files as FASTQ files are now available. They can be used by the researcher to detect genetic variants or genes related to disease severity or progression [4,5].

Genomic data such as methylation, gene expression or copy number alteration (CNA) matrices, from the same samples, can be analyzed individually or by integrating multiple domains at the same time. Examples of the first kind of approach include the application of deep learning methods to extract informative genes from gene expression profiles [6]. Among the approaches that process multiple domains at the same time, examples include the analysis of the association between gene expression and the presence of CNA events [7], or the integration of multiple domains by data fusion to cluster patients into groups with different survival [8]. Data fusion (DF) methods were developed to merge different data domains in an unsupervised way for feature selection and sample clustering [8,9,10]. DF applications such as joined singular value decomposition (jSVD) integrate the information from multiple domains to produce a single matrix; therefore, each sample is projected in a k dimensional space that can be used to define new clusters with methods such as k-means [8,9,11]. Other methods such as similarity network fusion (SNF) directly perform sample clustering based on the integration on one network of single-sample correlation matrices computed on each domain [12].

Uveal melanoma, a rare cancer of the eye that affects two to eight people a year per million people, has been molecularly characterized in great detail. Genomic analyses have shown that it is driven by a very limited number of driver events, and the analysis of gene expression, chromosome copy number alterations and DNA methylation concordantly reveals the existence of four major risk related subtypes that are clearly distinct from each other and tightly linked to the development of metastases and disease-free and overall survival after diagnosis [13]. UM has a very low mutational burden of 17–30 somatic mutations that affect protein coding sequences and might have functional consequences. Apparently, a single initiator mutation in one of four genes (GNAQ, GNA11, PCLB2 and CYSLTR2) is sufficient to yield a tumor that, upon acquisition of an additional mutation in the genes BAP1, a tumor suppressor gene, or SF3B1, a splice factor gene, and cytogenetic alterations, will progress to metastasis [13,14,15]. The four molecular classes are characterized by disomy of chromosome 3 without (class A) or with (class B) a hotspot SF3B1 mutation or monosomy of chromosome 3 and BAP1 mutation without (class C) or with (class D) amplification of chromosome 8q and an inflammatory infiltrate. Metastatic risk is low in class A, intermediate in class B and high in classes C and D [16]. These molecular classes are reflected by cytogenetic alterations and differential gene expression as well as differential DNA methylation [16,17].

Known molecular drivers and clearly distinct risk classes make UM especially suited for the development of data fusion approaches since it is straightforward to test classifications as to whether they improve classification over known classification systems based on single domains. However, it should be considered that genomics data are always characterized by some degree of noise from biological or technical factors (e.g., sample preparation, quality, etc.) and size limitations that prohibit perfect classification, which could instead be observed in an artificial training set [18].

Previously, we applied and adapted data fusion approaches to prognostic classification of UM. We performed joined singular value decomposition (jSVD) known in chemometrics as simultaneous component analysis, a simultaneous principal component analysis (PCA), and we developed joined constrained matrix factorization (jCMF) based on a form of coupled matrix factorization, also known as the k-table method, with a generalization of this factorization by allowing different constraints on the factor matrices [8,9].

Here, we report on the analysis of the uveal melanoma dataset with algorithms based on open-source code, i.e., R and Python implementations.

Information from multiple domains such as expression, methylation and copy number alteration from TCGA patients affected by UM were merged using data fusion or integration methods and applied to distribute samples in different clusters to perform feature selection, as we previously applied to the skin cutaneous melanoma TCGA dataset [9]. Different methods of data integration based on genomic domains are compared to evaluate which features (genes) are most relevant for UM subtypes and risk class detection and in which domains their effect is detectable (CNA level, gene expression or methylation). Selected methods will analyze genomic data that are relevant for UM subtyping in the high or low risk classes: CNA-based methods, which prioritize genes with expression levels altered by variation in copy number, data fusion or integration approaches will integrate expression and methylation data for patient clustering and feature selection as well as to identify which ones are transcriptionally predictive (i.e., genes with an association between methylation and expression levels).

## 2. Results

### 2.1. Gene Prioritization Methods

#### 2.1.1. Data Fusion

jSVD was used to integrate RNA-seq and methylation TCGA UVM data in order to produce a matrix U, with each patient defined by a three-dimensional space (Figure 1a,b). High- and low-risk classes are clearly separated in space (Figure 1a); the two clusters defined by k-means on the U matrix classified two patients as high risk who did not develop metastasis during follow-up in the blue cluster, which contains all low-risk samples (Figure 1b). The number of k-means clusters (2) was defined by balancing low-connectivity values while maximizing the silhouette score (Table 1).

At this point, we applied bootstrap analyses (significance analysis of microarrays, SAM [19]) to detect differentially expressed genes among the two classes of UM samples as defined by k-means. Samples of the two clusters are characterized by a set of differentially expressed genes and methylated probes (Figure 2). The two high-risk samples (class 3) that were clustered among the lower-risk cases by k-means show a methylation and expression profile that is more similar to their neighbors than the ones of the other group (in red, Figure 2). Generally, the classification as “high risk” of patients who did not develop metastases can be misclassification but must not be so, since they might develop metastases later on or have responded to therapy. The latter does not apply to UM for the absence of adjuvant therapy.

We tested the differentially expressed genes on a dataset of 253 UVM patients [17], using a multi-gene score (MGS); this produced two groups with a significant difference in terms of survival (Figure 3). Differentially methylated probes were tested only on the TCGA dataset; only one probe passed the multivariate testing (cg05522415, Appendix A).

#### 2.1.2. CNA Analysis Methods

The IGC R package (v 1.22) [7] was used to detect genes with expression values associated with CNA gain or loss. Considering a false discovery rate (FDR) below 0.05, 2036 genes were detected: 502 associated with CNA gain, the remaining 1534 with loss events. The CNAPE R package (https://github.com/WangLabHKUST/CNAPE, accessed on 24 January 2024) detects relevant features for CNA detection from RNA-seq data [20]; we used this package to develop a model able to distinguish between monosomic and disomic samples in chromosome 3, using TCGA UM data. A total of 299 genes were used to make the prediction and were considered for further analysis. These genes can distinguish disomic (low risk) from monosomic (high risk) patients.

#### 2.1.3. Methylation Analysis Methods

The MethylMix R package (v 2.32) [21] was used to detect genes with methylation levels associated with expression. This method uses a control group of samples to remove genes that are not differentially methylated compared to cancer samples and detect which ones are transcriptionally predictive (e.g., genes for which there is a significant inverse relationship of expression and methylation) [21]. Patients classified in class 2 by data fusion were considered as controls and those in class 1 were treated as tumor samples. Methylmix detected 90 genes as transcriptionally predictive.

### 2.2. Integration of Results

Several genes were detected using each method (Figure 4). In particular, data fusion selected 28 features that were also detected with CNAPE and IGC. CNAPE and IGC shared more genes compared to data fusion; this was expected since both methods are expected to detect genes with expression levels associated to CNAs, while data fusion analysis is based on RNAseq and methylation data. Among the seventeen DF selected genes that have passed survival analysis, two were detected using all methods (ROBO1, ROPN1, Table 2), while nine were shared with at least one R package based on CNA data analysis (IGC or CNAPE), and one by MethylMix.

ROBO1, ROPN1, BCHE, and CHL1 present lower gene expression in patients with a CNA loss at their locus (Figure 5): while ROBO1 and CHL1 gene expression reduced the score of the MGS signature, the opposite effect is produced by ROPN1 and BCHE (overexpressed in some samples with bad prognosis, as shown in Appendix A). ROBO1 and CHL1 map on chromosome 3p; the other two are located on 3q. CHL1 was found to be one of the most downregulated genes in UM that metastasized to the liver compared to non- metastatic tumors [22]. ROPN1 has previously been described as related to good prognosis when overexpressed in the UM TCGA dataset [23]; however, in the Piaggio et al. dataset [17], several metastatic patients have high expression levels of this gene (Appendix A). Among the genes selected by CNAPE and DF, we can find several genes related to worse prognosis. CADM1 and other genes involved in the production of cell adhesion molecules were found to be overexpressed in UM cells with BAP1 inactivation: experimental evidence supports a role of this gene in the metastasization process [24]. ITPR2 was previously described as mutated in the TCGA dataset; it is involved in G-protein-related pathways [15] and has been selected as part of a signature for tumor immune infiltration [25]. ISM1 was selected as a negative prognosis factor in a previously published 21-gene signature related to the UM tumor microenvironment, while MTUS1 and IL12RB2 were considered as indicators of favorable prognosis [26]. PDE4B was previously found as a protective factor in a prognostic signature based on inflammatory-related genes in UM [27]. ACSF2 was found to be among ferroptosis regulators in a signature, used to distinguish UM patients with different overall survival, that defined two clusters of patients with differences in prognosis and tumor-microenvironment-infiltrating cells [28,29]. CTF1 has been part of a previously defined UM-immune-related three-gene signature on TCGA data [30]. CARD11 was detected as a prognostic marker, with high expression associated to poor OS in the TCGA UVM dataset; in particular, metastatic patients had higher expression of this gene [31]; however, the MGS based on a larger dataset [17] assigned a protective effect to this gene, probably due to a set of patients with limited survival, metastatic disease and low CARD11 expression (Appendix A). HTR2B, TNFRSF19 and PTGER4 were previously found to be overexpressed in class 2 tumors (metastatic) [32]; in particular, TCGA UVM patients with high PTGER4 expression had worse survival [33]. Gene set enrichment analysis of MGS elements (Table 2) shows that these genes are involved in inflammatory (CARD11, PDE4B, TNFRSF19, HTR2B) and cell-motility-related biological processes (MTUS1, ROBO1, PTGER4, CHL1, HTR2B, PDE4B, ROPN1, Figure 6, Appendix A).

Eventually, we considered whether there was any overlap between the 90 genes detected using MethylMix with a consistent correlation between expression and methylation levels in the high-risk data fusion class (1) and not in the other class (2, Figure 4). SLC25A38 was detected using all methods; it maps chromosome 3p and is downregulated in metastatic UM patients; inactivation of this gene has been shown to promote distant metastasis in mouse models [34]. Other genes such as PLXNB1 and HLA-A were part of an immune gene signature used to define two risk classes, one of which had higher immune cell infiltration and lower survival in the TCGA UVM dataset [35]. CTF1 and RAPGEF3 were previously reported to be parts of gene signatures related to tumor microenvironment and immune system [26,30], with the first seen downregulated and methylated in BAP1 mutated samples [36]; PALMD was found to have low expression in metastatic UM tissues [37], and GSTA3 in low-survival patients [38].

## 3. Discussion

Integration of multiple genomics and phenotype data is gradually unveiling the complex molecular biology behind genetic diseases and cancers [39,40,41]. Data fusion has been previously applied as a tool to cluster patients or how to extract relevant features for disease prognosis by integrating data of several NGS, imaging and other clinically related datasets from the same group of samples [42,43,44]. The main limitations to the application of these approaches are batch effects, the curse of dimensionality that arises with genomic data and missing information or heterogeneity (data incompleteness) [43]. Regarding the first point, in each sequencing experiment, technical differences among replicates could mask or mimic biological variation; for example, different sequencing coverage among two groups of samples sequenced with RNAseq could potentially lead to the discovery of several false positives, as differentially expressed genes, if samples are not properly normalized [45]. The curse of dimensionality resides in the fact that in an NGS experiment, the number of features greatly exceeds the number of samples [46], which can easily result in model overfitting [47] and the inability to extract any relevant biological features or perform meaningful classification using the same model in a different dataset. Data fusion or integration methods can work on a full dataset or on a limited subset of genes, i.e., the most variable features [8,10,12,48]. In this way, most of the non-informative features are removed, reducing the required computational resources and the noise inside the dataset. In this work, we have shown that data fusion can potentially improve patient classification, as two patients previously classified by single domain analysis as high risk, but that had not developed metastasis during follow up, were classified with low- and intermediate-risk patients (Figure 2, on the left). However, it is not clear whether this classification could be efficient in a larger dataset since, to date, TCGA UVM is the only publicly available multidomain uveal melanoma dataset. However, promising results were obtained by applying DF on UM samples with expression data only and on TCGA samples for which mean gene methylation data were also available [49]. Interestingly, 9 out of the 17 DF detected genes that passed MGS were also detected using CNAPE or IGC; 2 of them were associated with a CNA loss (ROBO1, ROPN1, BCHE, CHL1). Interestingly, ROPN1 and BCHE, both mapping on chromosome 3q, have generally low expression levels in TCGA patients that developed metastasis during follow-up but not so in other UM datasets (Appendix A). One explanation could be that several patients from datasets other than the TCGA dataset could have a partial deletion on chromosome 3, not involving these two genes (Appendix A). Unfortunately, no CNA data are available for these patients. The use of multiple datasets to evaluate the method is essential to obtain an accurate estimate of the reliability of a classification method. Limited training set size, in the past, had determined the development of overfitted bioinformatic models that were not superior to a random predictor in the classification of new samples [50]. In this work, we could only test the performance of the genes selected by data fusion applications with a multi-gene score on a larger dataset. Some of these genes were also described in different works regarding UM [51], while two of them (CHL1 and IL12RB2) were also found to be hypermethylated with low expression in invasive malignant melanoma cells [52]; in particular, CHL1 is in an hypermethylated region on 3p in TCGA class 2 UMs [53].

Data fusion research should focus on new methods of data integration from multiple domains. Some genes could be affected by multiple genomic events that inactivate their expression (as from mutation, CNA and methylation domain). Single domain analysis failed to detect these genes as significantly altered in tumors, while the analysis of multiple domains could be a strong basis to distinguish between genes with a functional role in pathogenesis and those not causally involved markers.

## 4. Materials and Methods

### 4.1. jSVD Data Preparation and Analysis

TCGA methylation and RNA-seq data were downloaded from Broad GDAC Firehose (https://gdac.broadinstitute.org, accessed on 31 January 2023). RSEM gene expression counts were filtered from outliers by removing genes with less than 100 or more than 10^6 counts over all samples. RNAseq data normalization was based on the blind vst normalization function, as implemented in the DESeq2 R package (v 1.32.0) [54]. Feature reduction was performed by selecting the 1500 genes with the highest MAD for RNA-seq and the 1% most variable methylation probes; these data were used as input for the jSVD python script, as previously applied by Amaro and coauthors [9], setting the number of columns produced by the U matrix to 3. Patient clustering on the U matrix, produced by the jSVD, was based on the k-means method (complete agglomeration, Euclidean distance) from the R package ConsensusClusterPlus [55]; the number of cluster k was selected by minimizing connectivity and maximizing silhouette score, as computed by the clvalid R package [56]. Differentially expressed genes and methylated features, among patient clusters, were extracted with the significance analysis of microarray as implemented in R (Samr) [19]. Resulting DEGs and differentially methylated probes were analyzed with SPSS Statistics 20; in particular, multivariate Cox regression and multi-gene score analysis was computed on Piaggio et al.’s dataset [17], and the same analysis was conducted on the methylation probes of the TCGA UVM dataset [16].

### 4.2. CNAPE and IGC

RNA-seq and CNA data analyzed using CNAPE and the IGC R package [7,20] were downloaded from cBioPortal (https://www.cbioportal.org/, accessed on 24 January 2024) [57,58]; only genes with a CNA in at least 4 samples were considered for further analysis. These pieces of software work on expression and CNA matrices with the same genes, as rows, and patients, as columns. IGC tests whether the expression of one gene is associated to CNA events overlapping the locus: detected relations could be “loss” if a decrease in RNA expression is associated with deletion events, “gain” if increased expression is associated with augmented copies of one gene, or “both” when the two events (gain and loss) are observed in the same gene [7]. In a first step, samples with CNA on one gene are classified as CNA-gain (“gain”, with an increase in CNA), CNA-loss (“loss”, with a decrease in CNA) or CNA-neutral (no CNA detected). At this point, a gene can be classified as gain or loss on the proportion of samples that have the CNA event (e.g., if more than 20% of samples have a CNA gain on that gene, it is classified as “gain”). As a final step, Student’s *t*-test with unequal variance is computed on the expression values. For each gene, a false discovery rate (FDR) and p value is reported; in this work, only “gain” and “loss” elements with an FDR below 0.05 were considered (as obtained with the find_cna_driven_gene function with standard parameters: gain, loss_prop = 0.2). CNAPE uses RNA-seq data to develop a model able to distinguish between samples with or without a large CNA event [20]. In this work, the model was trained on NGS data in order to distinguish between chromosome 3 gain or loss; the genes selected by the model to make a prediction were considered for further analysis and reported in Appendix A and Figure 4. The model was trained with the cv.glmnet function with default parameters, except the number of cross-validation folds, which was set to 20 to have stable results (md = cv.glmnet(x = as.matrix(dtx), y = dty, family =“binomial”, nfolds = 20, alpha = 0.1))

### 4.3. MethylMix

RNA-seq UVM data were downloaded from cBioPortal [57,58], and mean gene methylation levels were obtained from https://gdac.broadinstitute.org/ (accessed on 31 January 2023). The table of mean gene methylation was split in two, the first composed of samples classified in class 1 by data fusion and considered as cancer samples (*METcancer*), the second comprising class 2 patients, treated as control samples (*METnormal*). RNA-seq data of class 1 patients were retrieved from cBioportal normalized expression data and treated as a cancer gene expression profile (*MAcancer*). The Methylmix R package [21] was used to detect transcriptionally predictive genes with the *MethylMix* function *MethylMix(METcancer, GEcancer, METnormal)*. Briefly, genes with different methylation levels in cancer and control data were tested to assess whether they had a significant relationship with expression data.

### 4.4. Joint Singular Value Decomposition

Joint singular value decomposition, described in [8,59], was developed with the Python package Pymanopt (v 0.2.5) [60]. jSVD factorizes each genomic data matrix *A* as (1):(1)$$A\approx U\Sigma_{i}V_{i}^{T}$$

$\Sigma_{i}$ is a singular value diagonal matrix; the others are orthonormal. The *U* matrix is shared among each matrix decomposition; therefore, it represents the fused information from *A* datasets and is used for patient clustering. A Riemannian Trust scheme has been used to obtain a minimum on the product of Stiefel manifolds (set as Product([Stiefel(I,k), Stiefel(N1,k), Stiefel(N2,k)]: N1 and N2 represent the number of genes or methylation probes of the RNA-seq, Methylation matrix, respectively). The minimization was stopped when the norm of the projected gradient was lower than $1^{-12}$ (mingradnorm = 1 × 10^−12^).

### 4.5. Gene Signature Performance Evaluation

Features selected by all methods presented in this paper (CNAPE, IGC, MethylMix and Data Fusion) were assessed as gene signatures to predict chromosome 3 monosomy and metastatic disease development compared to chromosome 3 monosomy on the Piaggio et al. 2022 dataset [17] in terms of AUC, as previously applied for signature and phenotype prediction validation [34,61,62,63]. Gene signature scores were computed with the *simpleScore()* function of the signscore R package [64,65]. The computed TotalScore of each signature and overall score computed on available genes reported in Appendix A were converted to a value between 0 and 1 by subtracting to each value the minimum value of the signature and dividing it by the difference of the maximum and minimum value of the signature. The ROCR package (v 1.0-11) [66] was used to compute the ROC curves and relative AUC of each signature, by comparing the difference between 1 and the signature score (except for IGC gain) and the M3 or metastasis classes.

## 5. Conclusions

In this work, different data integration and single domain gene prioritization methods were applied to the UM TCGA dataset. Most of the genes detected using IGC and CNAPE are located on chromosomal positions where relevant CNAs used for clinical assessment of UM metastatic risk are present (1p, 16q, 3p loss and 6p, 8q gain, Table 3) [67].

Chromosome 16q and 1p deletion were found to increase metastatic risk in patients with M3 and chromosome 8 amplification [67]. IGC prioritizes CNA associated genes on the basis of related RNA expression. Therefore, genes that are not strictly regulated by deletion or gain events will not be detected. CNAPE selected a set of genes able to discriminate between chromosome 3 monosomic and disomic patients of the TCGA UVM dataset: the genes detected were not only localized on 3p or 3q, since features in other genomics locations were used for M3 prediction. Data-fusion- and MethylMix-prioritized feature localization was more dispersed on the whole genome compared to genes detected using IGC: the integration of RNAseq and methylation array data also prioritizes genes that are not strictly regulated by CNAs. Therefore, the integration of results from different gene selection methods can detect features that are relevant for UM prognosis but are not detectable in a single genomic domain. The prediction performance of the signature detected using all methods presented in this work (Appendix A) is reported in Figure 7. In general, gene expression signatures predictive of chromosome 3 monosomy obtained higher AUC values compared to metastatic disease onset estimation. This reflects the fact that chr3 monosomy is certain at the time of analysis whereas metastases can also develop after the end of follow-up. High-risk cases that did not develop metastases during follow-up might do so afterwards.

Interestingly, CNAPE outperformed all methods on M3 prediction, obtaining high AUC in the TCGA and the remaining part of the Piaggio dataset [17] (Figure 7a,b). If we compare the performance of IGC loss on M3 prediction, we can observe a consistent decrease in the AUC score by comparing the dataset where the gene signature was computed and a different test set (AUC from 0.91 to 0.77). It should be taken into account that IGC simply detects genes with expression level associated to CNA of the gene, without performing any supervised feature selection for M3 prediction. Therefore, worse performance in different datasets could be expected. Regarding the prediction of metastatic disease from gene signatures, a general decrease in the performance is observable comparing the dataset where features were extracted with the validation datasets (Figure 7c,d). Considering only the TCGA dataset, some methods show a performance superior to chromosome 3 classification for metastatic risk prediction, while all curves are near M3 classification in all other samples (Figure 7d).

## Figures and Tables

**Figure 1 ijms-25-01796-f001:**
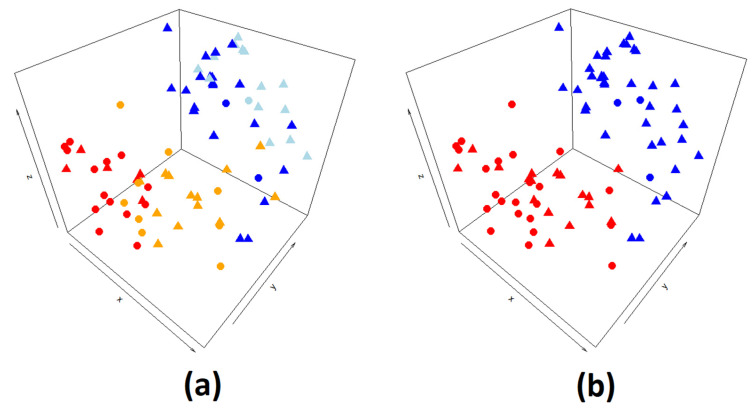
(**a**) Scatterplot of the 80 patients from the UVM TCGA dataset in the three-dimensional (k = 3) U matrix produced using jSVD data integration of RNA-seq and methylation array data. Points are colored according to the metastatic risk classes [16], from high (4) to low (1): 4 in red, 3 in orange, 2 in blue, 1 in azure. Patients that developed metastasis are reported as circles. (**b**) Scatterplot of the 80 patients from the UVM TCGA dataset in the three-dimensional (k = 3) U matrix produced using jSVD data integration of RNA-seq and methylation array data. Points are colored according to the two clusters defined by k-means on the jSVD U matrix.

**Figure 2 ijms-25-01796-f002:**
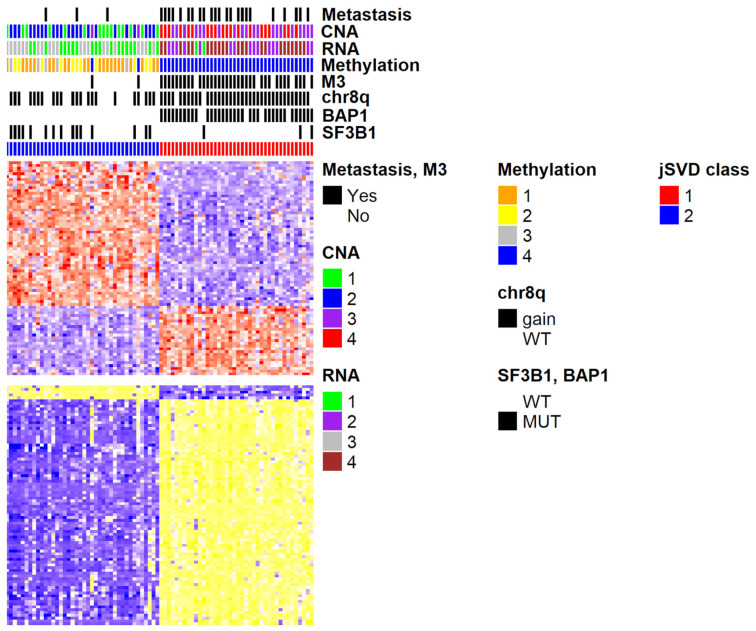
Heatmap of the differentially expressed genes and methylated probes of the 80 patients from the UVM TCGA dataset, considering the two clusters detected on the jSVD U matrix. At the top of the figure, each row represents different sample features, respectively: presence of metastasis, CNA metastatic risk, RNA and methylation cluster class [16], loss of chromosome 3 (M3), gain of chromosome 8q, mutations on BAP1 and SF3B1.

**Figure 3 ijms-25-01796-f003:**
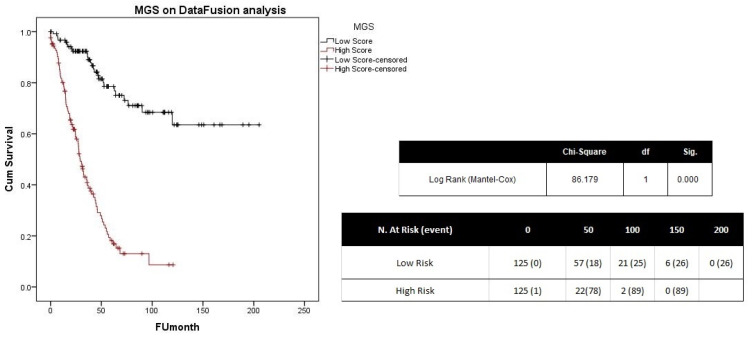
KM curves of patients with a high or low MGS considering the differentially expressed genes detected using SAM analysis on the two jSVD-related clusters. Low score curve is in black, while the high one is reported in red.

**Figure 4 ijms-25-01796-f004:**
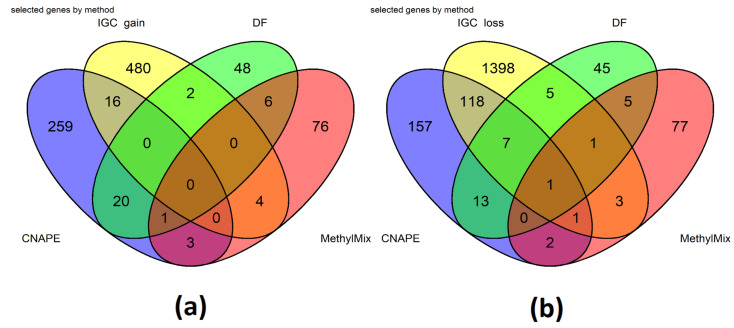
(**a**,**b**) Genes detected using different data analysis and integration methods: most of the data fusion genes were not detected using any other methods (48, 43 also considering IGC gain and MethylMix), 13 are shared with CNAPE (**b**), and 8 were detected using CNAPE and IGC as low expression driven by CNA loss; 2 were also detected as CNA gain by IGC; 5 are only shared with IGC loss (**b**). Only 7 genes are shared between DF and Methylmix (**a**,**b**).

**Figure 5 ijms-25-01796-f005:**
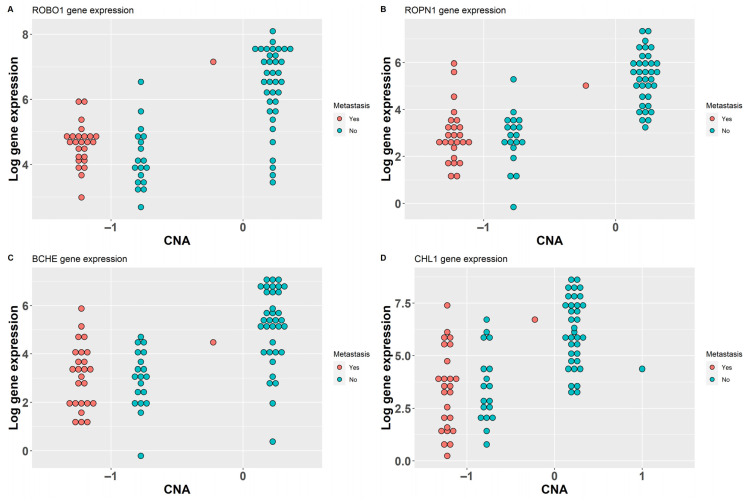
Expression level and CNA state of genes detected using data fusion and IGC (TCGA UVM dataset): low level of expression is associated with CNA loss. The Y axis reports the log expression values, while the x axis reports the CNA state with −1 as loss, 1 as gain and 0 as neutral (e.g., no CNA). Metastatic samples have generally low expression values compared to copy neutral (0) samples. Each panel represents the expression levels of one gene in the TCGA dataset: ROBO1 (**A**), ROPN1 (**B**), BCHE (**C**), CHL1 (**D**).

**Figure 6 ijms-25-01796-f006:**
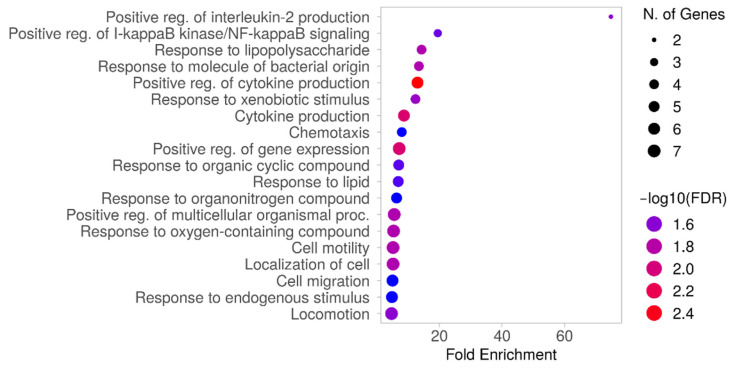
Gene set enrichment analysis on MGS features (Table 2, FDR cutoff of 0.05. Enriched GO BP terms are reported ordered on the basis of the number of genes of the signature in Table 2 and fold enrichment. Dot size is based on the number of genes (in Table 2) involved in the process; color is based on FDR.

**Figure 7 ijms-25-01796-f007:**
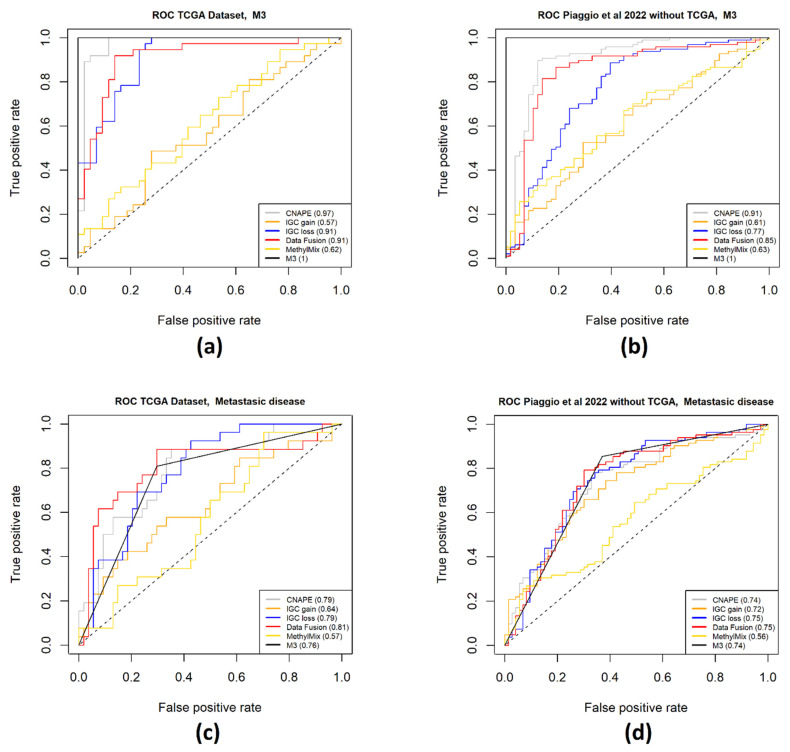
ROC curves of gene-signature-based prediction of M3 and metastatic disease development during follow up. Each ROC curve was computed on the TCGA UVM (**a**,**c**) and the Piaggio dataset [17] (**b**,**d**). The four panels report the performance of gene signatures on M3 (**a**,**b**) and metastasis prediction (**c**,**d**). AUC scores are reported in each panel legend; chromosome 3 monosomy ROC curve is reported as a black line and can be used as a reference for comparison.

**Table 1 ijms-25-01796-t001:** Performance measures used to define the optimal number of clusters (k). Connectivity is 0 if one sample has no neighbors from different clusters, while silhouette score represents sample fit in its cluster.

Number of k	2	3	4
Connectivity score	2.25	6.27	15.66
Silhouette score	0.45	0.51	0.52

**Table 2 ijms-25-01796-t002:** Data fusion genes selected using multi-gene score (MGS) procedure. Genes are ordered considering the number of different methods that detected the gene (n overlap column). Column 1 to 3 report presence (1) or absence (0) of the gene using each method: CNAPE and IGC are CNA loss and data fusion, respectively. The cytoband and the multi-gene score (MGS score) of each gene are reported in the last columns. Of all genes in Table 2, only the CTF1 gene was also detected using MethylMix.

GENE	CNAPE	IGC Loss	Data Fusion	n Overlap	Cytoband	MGS Score
ROBO1	1	1	1	3	3p12.3	−0.241
ROPN1	1	1	1	3	3q21.1	0.312
CADM1	1	0	1	2	11q23.3	0.233
ITPR2	1	0	1	2	12p12.1	−0.323
ISM1	1	0	1	2	20p12.1	0.213
PDE4B	1	0	1	2	1p31.3	−0.291
ACSF2	1	0	1	2	17q21.33	0.302
BCHE	0	1	1	2	3q26.1	0.274
CHL1	0	1	1	2	3p26.3	−0.152
IL12RB2	0	0	1	1	1p31.3	−0.225
MTUS1	0	0	1	1	8p22	−0.276
CTF1	0	0	1	1	16p11.2	−0.301
CPS1	0	0	1	1	2q34	0.177
HTR2B	0	0	1	1	2q37.1	0.21
CARD11	0	0	1	1	7p22.2	−0.254
TNFRSF19	0	0	1	1	13q12.12	0.125
PTGER4	0	0	1	1	5p13.1	0.12

**Table 3 ijms-25-01796-t003:** Chromosome localization of genes reported in Appendix A. For each method, the number of genes mapping on chromosomes relevant for cytogenetic characterization of uveal melanoma are reported.

Chr	CNAPE	IGC Gain	IGC Loss	Data Fusion	MethylMix
1p	9	0	606	5	6
1q	8	0	0	1	4
3p	71	0	301	7	4
3q	53	0	285	5	0
6p	11	265	0	2	5
6q	3	0	244	1	2
8p	5	1	0	2	1
8q	11	236	0	3	3
16p	3	0	0	2	2
16q	2	0	98	1	0
other	123	0	0	48	63
total	299	502	1534	77	90

## Data Availability

Data used in this work for data integration can be downloaded from http://cancergenome.nih.gov/ and cBioportal, as described in the Material and Methods section. The UVM gene expression test set, described in [17], is composed of public datasets (GSE27831, GSE51880, TCGA-UVM) and the Leiden dataset, which is available upon request to authors [68].

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
