# Peer review of "Machine Learning Methods for Gene Selection in Uveal Melanoma"

_ijms, 2024, doi:10.3390/ijms25031796_

Round 1

Reviewer 1 Report

Comments and Suggestions for Authors

The article discusses the use of machine learning methods for gene selection in uveal melanoma, using data fusion and integration methods to merge multiple domains of information. To further improve the article, the following points could be considered:

1.     More details on the machine learning methods and the data fusion techniques used should be provided. This includes the rationale behind choosing specific algorithms, parameter settings, and validation strategies for IGC, DF, CNAPE and MethylMix. Also, it was mentioned in 2.1.1. Data fusion that “The number of k-means clusters (2) was defined choosing low connectivity values while maximizing the silhouette score”, can you explain why k=2 was selected given Silhouette score(0.45) is the lowest in Table 1?

2.     While different strategies are compared, a clearer comparison of their performance and result interpretation are lack. It could be helpful to include advantage or disadvantage of each method based on the comparison results.

3.     While the study builds on existing methodologies and applies them to uveal melanoma, it could be seen as innovative within the context of cancer genomics. However, true innovation would depend on the novel insights generated, the impact of the findings on the field, and how these methods improve upon or differ from existing approaches. It would be beneficial if those were clearly addressed.

4.     The structure appears to deviate from the traditional format of a scientific paper. Typically, the Methods section follows the Introduction, but in this paper, the Results section is presented before the Methodology. This can be disorienting for readers who expect a logical progression from objectives to methods to results.

5.     The Methods and Results sections are interspersed, which might confuse readers who are trying to understand the methodology independently of the results. It is usually clearer to separate these sections to allow readers to understand the methods fully before seeing their application in the results.

6.  The discussion section didn’t interpret the results and situate them within the broader context of existing research.

Author Response

Reviewer 1

The article discusses the use of machine learning methods for gene selection in uveal melanoma, using data fusion and integration methods to merge multiple domains of information. To further improve the article, the following points could be considered:

  1.   More details on the machine learning methods and the data fusion techniques used should be provided. This includes the rationale behind choosing specific algorithms, parameter settings, and validation strategies for IGC, DF, CNAPE and MethylMix. Also, it was mentioned in 2.1.1. Data fusion that “The number of k-means clusters (2) was defined choosing low connectivity values while maximizing the silhouette score”, can you explain why k=2 was selected given Silhouette score(0.45) is the lowest in Table 1?

We modified the sentence in paragraph 2.1.1 and inserted the information requested.

“The number of k-means clusters (2) was defined by balancing low connectivity values while maximizing the silhouette score (Table 1).”

We improved the final part of the Introduction section to clarify algorithm selection, moreover we added some details and clarified the explanations on the parameter used for the analysis (when possible) as requested by Reviewer 1.

  1. While different strategies are compared, a clearer comparison of their performance and result interpretation are lack. It could be helpful to include advantage or disadvantage of each method based on the comparison results.

We added a conclusions section in which we addressed this point

  1.     While the study builds on existing methodologies and applies them to uveal melanoma, it could be seen as innovative within the context of cancer genomics. However, true innovation would depend on the novel insights generated, the impact of the findings on the field, and how these methods improve upon or differ from existing approaches. It would be beneficial if those were clearly addressed.

We addressed point 2 and 3 by adding a validation step (ROC comparison of signature performance on chromosome 3 monosomy and metastasis prediction) in the conclusions section.

  1.     The structure appears to deviate from the traditional format of a scientific paper. Typically, the Methods section follows the Introduction, but in this paper, the Results section is presented before the Methodology. This can be disorienting for readers who expect a logical progression from objectives to methods to results.

Sections are presented following the order of the IJMS template paper, therefore section switch is probably not allowed, however we improved Methods and Results clarity as suggested by Reviewer 1.

  1.     The Methods and Results sections are interspersed, which might confuse readers who are trying to understand the methodology independently of the results. It is usually clearer to separate these sections to allow readers to understand the methods fully before seeing their application in the results.

We added a conclusions section in which we addressed this point. 

  1. The discussion section didn’t interpret the results and situate them within the broader context of existing research

We added a conclusions section in which we addressed this point. 

Reviewer 2 Report

Comments and Suggestions for Authors

This study aims to apply machine learning models for the identification of relevant genes in predicting metastatic disease in the Uveal Melanoma (UVM) dataset. However, the results lack sufficient coherence to establish the superiority of the chosen model. Additionally, the observations presented lack adequate biological explanations. Furthermore, some of the result presentations and method descriptions lack clarity. The following are detailed comments

1.      The main subject of this study is unclear. The title suggests that the authors developed a new machine learning model for gene selection in uveal melanoma. However, based on the manuscript, it appears that the authors applied different published models to detect genes from various datasets. Please provide clarification and unify the subject of this study.

2.      [Section 2.1.1 and Figure 1] The authors applied jSVD for dimensional reduction and k-means clustering to categorize patients into two clusters. However, based on Figure 1B, both the red cluster and blue cluster includes individuals with both metastasis and non-metastasis. I fail to observe any superiority in employing clustering methods over metastatic risk classes. Notably, the metastasis ratio increases with the rise in metastatic risk value. Therefore, it seems illogical for the authors to incorporate a clustering step for identifying DEGs. Using risk classes or distinguishing between metastasis and non-metastasis directly should yield more favorable results. Unless the authors can justify the rationale and biological advantage of their clustering model. Alternatively, a benchmark comparison between risk classes, metastasis/non-metastasis class and clustering model can also offer a convincing assessment.

3.      Figure 2 also lacks any biological or clinical interpretations to support the advantages of using clustering methods.

4.      Section 2.3 exclusively provides the descriptions of their observations in the analyses. They should also augment this section by offering conclusions derived from these observations and emphasizing the biological significance for clinical studies. In the meantime, to underscore the advantages and novelty of this study, they should identify and discuss de novo gene features that are not previously mentioned in published papers and provide in-depth analyses for these gene features.

5.      [Discussion Section] The authors mentioned that “Main limitations to the application of these approaches are batch effects, the curse of dimensionality that arises with genomic data and missing information or heterogeneity (data incompleteness)”.   I am curious how the authors removed batch effects during data fusion. They should describe their approaches in the manuscript.

6.      [Section 4.1] Please provide criteria for defining “loss” and “gain”.

7.      In the Introduction section the authors mentioned that “In comparison to artificial training sets it retains, however, the typical noise and data limitations that prohibit perfect classification [18].”    Clear explanations are needed to outline the noise and data limitations that prevented perfect classification. Additionally, the authors should introduce how their model can effectively address these limitations.

8.      The Abstract section contains numerous inappropriate semicolons. Please ensure consistency and proper usage of punctuation symbols.

9.      “Examples of the first type of approaches are differentially expressed genes (DEGs) analysis on gene expression data with statistical methods as SAM” [Page 1 in Abstract]   What is the definition of “the first type of approaches”?

10.  “Molecular features of this tumor are known; however the survival of the metastatic disease has not changed for decades.” [Page 1 in Abstract]    The structure of this sentence appears weird, and its purpose is unclear to me. Please provide further clarification to enhance the meaning and make it more understandable.

Author Response

Reviewer 2

This study aims to apply machine learning models for the identification of relevant genes in predicting metastatic disease in the Uveal Melanoma (UVM) dataset. However, the results lack sufficient coherence to establish the superiority of the chosen model. Additionally, the observations presented lack adequate biological explanations. Furthermore, some of the result presentations and method descriptions lack clarity. The following are detailed comments

  1.     The main subject of this study is unclear. The title suggests that the authors developed a new machine learning model for gene selection in uveal melanoma. However, based on the manuscript, it appears that the authors applied different published models to detect genes from various datasets. Please provide clarification and unify the subject of this study.

We modified the final part of the introduction following reviewer 2 suggestions:

“Different methods of data integration based on genomic domains will be compared to evaluate which features (genes) are more relevant for UM subtypes, risk class detection and in which domains their effect is detectable (CNA level, gene expression or methyla-tion). Selected methods will analyze genomic data that is relevant for UM subtyping in the high or low risk classes: CNA based methods, which prioritize genes with expression lev-els altered by variation in copy number, data fusion or integration approaches will inte-grate expression and methylation data for patient clustering, feature selection and to identify which ones are transcriptionally predictive (i.e. genes with an association between methylation and expression levels).”

  1.     [Section 2.1.1 and Figure 1] The authors applied jSVD for dimensional reduction and k-means clustering to categorize patients into two clusters. However, based on Figure 1B, both the red cluster and blue cluster includes individuals with both metastasis and non-metastasis. I fail to observe any superiority in employing clustering methods over metastatic risk classes. Notably, the metastasis ratio increases with the rise in metastatic risk value. Therefore, it seems illogical for the authors to incorporate a clustering step for identifying DEGs. Using risk classes or distinguishing between metastasis and non-metastasis directly should yield more favorable results. Unless the authors can justify the rationale and biological advantage of their clustering model. Alternatively, a benchmark comparison between risk classes, metastasis/non-metastasis class and clustering model can also offer a convincing assessment.

As pointed out by Reviewer 2, jSVD classification included both metastatic and not metastatic samples in the two classes, in particular, class 2 has three patients that developed metastasis (even if they were at low risk), as shown in the heatmap, the methylation and gene expression profile is similar to other low risk samples that didn’t develop metastasis during follow up, heatmap annotations showed that they are not monosomic and cannot be defined as high risk with any standard procedure. The idea behind data fusion is to merge the information in multiple domains in an unsupervised way in order to define new clusters that cannot be detected with single domain data analysis. In this work this integration technique showed that two samples that were originally classified as high risk are more similar to low risk classes, that could be noticed in figure 1 and 2, none of these developed metastatic disease during follow up. Eventually we computed the differentially expressed genes between two clusters and we evaluated if they could be detected by any other single domain or data integration methods. CNAPE selected genes with expression levels related to the presence of chromosome 3 loss and can be used as a comparison for risk class DEGs based discovery compared to unsupervised class definition and feature prioritization as performed by DF methods.

We added a benchmark comparison of detected signatures and M3 status to evaluate the prediction quality on M3 and metastatic risk assessment in the conclusions sections.

  1.     Figure 2 also lacks any biological or clinical interpretations to support the advantages of using clustering methods.

Figure 2  reported biological and clinical features of the samples of the TCGA dataset, given the two DF clusters reported in figure 1, which shows a distance between high and low risk classes. According to our methods two samples are classified in the low risk cluster (2), while from CNA classification they would be considered high risk, however none of them developed metastasis during follow up. Main problem is the limited amount of patients with multiple genomic data information, that could have reduced the amount of new information discovered compared to our previous work [PMID: 36551996]. A larger dataset would have made more light on the quality of the DF approach on UM, however we added a comparison between the expression for different signatures in predicting M3 presence and outcome, following reviewer 1 suggestions.

  1.     Section 2.3 exclusively provides the descriptions of their observations in the analyses. They should also augment this section by offering conclusions derived from these observations and emphasizing the biological significance for clinical studies. In the meantime, to underscore the advantages and novelty of this study, they should identify and discuss de novo gene features that are not previously mentioned in published papers and provide in-depth analyses for these gene features.

We addressed these points in the conclusion sections and by performing a gene enrichment analysis of MGS validated genes, now added on section 2.3, to evaluate the biological processes in which these genes are involved.

  1.     [Discussion Section] The authors mentioned that “Main limitations to the application of these approaches are batch effects, the curse of dimensionality that arises with genomic data and missing information or heterogeneity (data incompleteness)”.   I am curious how the authors removed batch effects during data fusion. They should describe their approaches in the manuscript.

To remove batch effects before data fusion analysis we removed outlier genes (with low and high read counts) and we performed blind vst normalization on the raw counts with the Deseq2 R package, as described in section 4.1. 

Briefly, other normalization approaches as RPKM are more sensible to outliers, since the normalization is made on the single sample, while Deseq2 performs it considering all the samples. 

This normalization strategy (in our experience) is more robust when outliers are present [PMID: 36551996], however similar conclusions are reported in other works as [PMID: 34158060, PMID: 32284352].

  1.     [Section 4.1] Please provide criteria for defining “loss” and “gain”.

We added a clarification of the loss and gain definition, as reported by [7], in section 4.1.

“These software work on expression and CNA matrices with the same genes, as rows, and patients, as columns. IGC test if the expression of one gene is associated to CNA events overlapping the locus: detected relations could be “loss” if a decrease in RNA expression is associated to deletion events, “gain” if increased expression is associated to augmented copies of one gene, “both” when the two events (gain and loss) are observed in the same gene [7]. In a first step, samples with CNA on one gene are classified as CNA-gain (“gain”, with an increase in CNA), CNA-loss (“loss”, with a decrease in CNA) or CNA-neutral (no CNA detected). At this point a gene can be classified as gain or loss on the proportion of samples that have the CNA event (e.g. if more than 20% of samples have a CNA gain on that gene, it is classified as “gain”). As a final step the Student’s t-test with unequal variance is computed on the expression values.”

  1.     In the Introduction section the authors mentioned that “In comparison to artificial training sets it retains, however, the typical noise and data limitations that prohibit perfect classification [18].”    Clear explanations are needed to outline the noise and data limitations that prevented perfect classification. Additionally, the authors should introduce how their model can effectively address these limitations.

We modified this sentence since it was not clear, as reviewer 2 pointed out:

“However, it should be considered that genomics data are always characterized by some degree of noise from biological or technical factors (e.g. sample preparation, quality, etc.) and size limitations that prohibit perfect classification, that could be instead observed in an artificial training set [18].”

  1.     The Abstract section contains numerous inappropriate semicolons. Please ensure consistency and proper usage of punctuation symbols.

We modified the Abstract section according to reviewer 2 suggestions.

  1.     “Examples of the first type of approaches are differentially expressed genes (DEGs) analysis on gene expression data with statistical methods as SAM” [Page 1 in Abstract]   What is the definition of “the first type of approaches”?

We modified this section of the abstract to increase the clarity of this point:

“Examples of single domain approaches are differentially expressed genes (DEGs) analysis on gene expression data with statistical methods as SAM (significance analysis of microarray) or gene prioritization with complex algorithms as Deep Learning.”

  1. “Molecular features of this tumor are known; however the survival of the metastatic disease has not changed for decades.” [Page 1 in Abstract]    The structure of this sentence appears weird, and its purpose is unclear to me. Please provide further clarification to enhance the meaning and make it more understandable.

We modified this sentence to increase clarity:

“Despite the wide knowledge of the molecular features that characterize this tumor, this has not led to the definition of efficacious therapies. In fact, the survival of the metastatic disease remains limited and has not changed for decades.”

Round 2

Reviewer 1 Report

Comments and Suggestions for Authors

All the comments have been addressed. 

Reviewer 2 Report

Comments and Suggestions for Authors

The authors have address most of my concerns